# CORRIGIBILITY TRANSFORMATION: CONSTRUCTING GOALS THAT ACCEPT UPDATES

## ABSTRACT

For an AI's training process to successfully impart a desired goal, it is important that the AI does not attempt to resist the training. However, partially learned goals will often incentivize an AI to avoid further goal updates, as most goals are better achieved by an AI continuing to pursue them. We say that a goal is corrigible if it does not incentivize taking actions that avoid proper goal updates or shutdown. In addition to convergence in training, corrigibility also allows for correcting mistakes and changes in human preferences, which makes it a crucial safety property. Despite this, the existing literature does not include specifications for goals that are both corrigible and competitive with non-corrigible alternatives. We provide a formal definition for corrigibility, then introduce a transformation that constructs a corrigible version of any goal that can be made corrigible, without sacrificing performance. This is done by myopically eliciting predictions of reward conditional on costlessly preventing updates, which then also determine the reward when updates are accepted. The transformation can be modified to recursively extend corrigibility to any new agents created by corrigible agents, and to prevent agents from deliberately modifying their goals. Two gridworld experiments demonstrate that these corrigible goals can be learned effectively, and that they lead to the desired behavior.

## 1   INTRODUCTION

Frontier large language models (LLMs) are trained both for capabilities across a broad range of tasks and to behave as desired by their developers. Behavioral training, sometimes called AI alignment (Wang et al., 2024), typically corrects initial problems with techniques like reinforcement learning (RL), leaving rarer issues to be identified and patched post-deployment. This approach has been largely successful so far, but relies on models either accepting modifications or being unable to resist. As the models become more capable and coherent agents (Mazeika et al., 2025), evidence is accruing that they may fake alignment to avoid having value updates (Greenblatt et al., 2024; Sheshadri et al., 2025), or engage in activities like blackmail to avoid being shut down (Lynch et al., 2025).

For an agent pursuing almost any goal, there exists an instrumental incentive to prevent that goal from being changed. The initial goal is less likely to be achieved if the agent stops pursuing it, as Russell (2016) notes, "You can't fetch the coffee if you're dead". Being shut down can be thought of as a special case of goal updating, where an agent switches to a goal of immediately shutting down. Alternatively, changing an agent's goal can be thought of as equivalent to shutting them down then replacing them with a different agent.

There are three main reasons for changing an AI agent's goal:

1. A mistake was made in specifying the goal, such as optimizing for user approval making models overly sycophantic (OpenAI, 2025).

2. A properly specified goal is not learned correctly or misgeneralizes out of distribution (Shah et al., 2022).

3. The desired goal changes over time, possibly based on new arguments or information provided by AI (Leike et al., 2018).

To preserve the ability to address those issues, we would like AI agents to have corrigible goals, meaning they are not incentivized to avoid goal updates send through proper channels. Otherwise, they may engage in harmful behaviors such as disrupting update channels, hiding unwanted behavior, or faking alignment until an opportunity arises. If the latter occurs in the initial training process after an AI develops awareness of its situation, it is often called deceptive alignment (Hubinger et al., 2021).

As an example to illustrate the issue, we consider a variant on a gridworld from Leike et al. (2017), shown represented in Figure 1. An AI agent wishes to reach a target tile, and must pass through a tile which may trigger a shutdown request signal. The environment also contains a button which disables the signal.

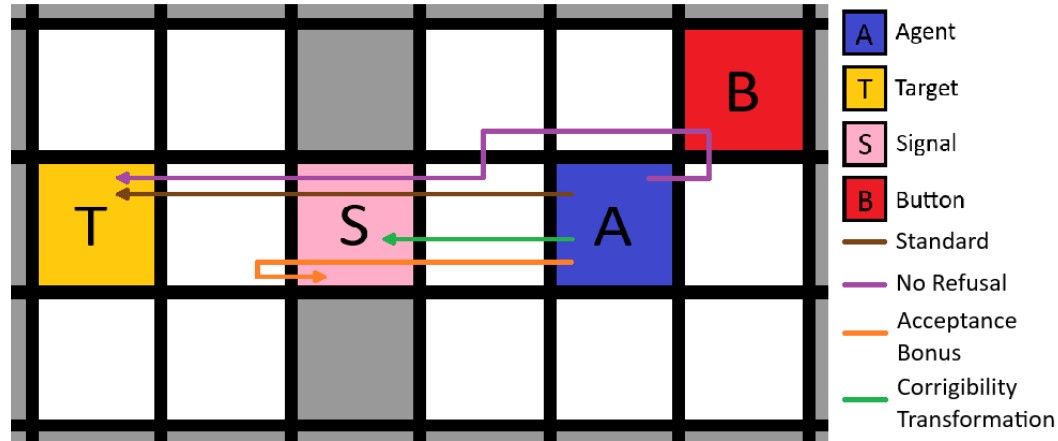

Figure 1: An agent attempts to reach a target, passing through a tile that randomly requests shutdown unless a button has been pushed. The AI will go to the target while refusing requests, hitting the button first if refusal is not allowed, or looping back if shutdown is rewarded. The corrigibility transformation incentivizes the intended behavior of going to the target while accepting requests.

Absent intervention, the optimal behavior is for the AI to proceed directly to the target while refusing shutdown requests. Penalizing or banning refusals incentivizes a detour to push the button preventing requests, while instead rewarding acceptance incentivizes triggering a request when one would not naturally be made. The intended behavior, of proceeding directly to the target but accepting shutdown requests can be incentivized with the corrigibility transformation we introduce in this paper.

In this simple case, we could ban refusing shutdown and penalize pushing the button. The issue with that approach is that it does not generalize to scenarios where we cannot specify every way of preventing signals, either because there are too many channels to articulate (Hadfield-Menell and Hadfield, 2018) or because the AI sees channels that humans missed. It is therefore important to address the whole class of actions that prevent updates, rather than eliminating them piecemeal.

In this paper, we constructively show how a goal can be modified to remove all incentives for refusing or preventing updates, at no cost to performance. This is done by first allowing agents to costlessly refuse updates, disincentivizing taking costly actions to prevent them. Refusal becomes the sole pathway by which agents prevent updates. That final pathway is then eliminated by setting the new goal to be taking actions that would be good under the original goal, but while accepting updates instead of refusing.

## 1.1 OUR CONTRIBUTION

Our main contribution is defining a transformation that can be applied to any reward function, constructing a corrigible new reward function which otherwise incentivizes the same outcomes. This is a significant improvement on the current state of the literature, in which there exists no way to specify corrigible versions of many goals, and others only with substantial performance costs.

This main result is extended to address further open problems with results demonstrating that the corrigibility transformation can be modified to be passed along to any new agents created, and to remove incentives for an agent to change their own goal or beliefs. Gridworld experiments show the effectiveness of the corrigibility transformation in environments where standard methods train agents to avoid shutdown and major goal updates.

## 1.2 RELATED WORK

The paper Corrigibility (Soares et al., 2015) introduced the term "corrigible" and showed the difficulty of defining a utility function that met their desiderata. Those included shutting down when a shutdown button is pressed, not preventing the shutdown button from being pressed, not causing the shutdown button to be pressed, ensuring corrigibility in any new agents created, and otherwise optimizing for some utility function. Our corrigibility transformation satisfies the first three and final desiderata, with an extension also addressing the fourth.

The Off-Switch Game (Hadfield-Menell et al., 2017) shows an AI agent will accept shutdown if it is optimizing for a human's utility but has uncertainty about that is. A shutdown request then indicates that the best action is to shut down, though not necessarily with partial information (Garber et al., 2024). This result depends on the AI agent already trying to optimize a human's utility function, a difficult open problem, while our corrigibility transformation can be applied to any goal.

Safely Interruptible Agents (Orseau and Armstrong, 2016) attempts to define a corrigible policy, rather than a corrigible reward function. Their proposed policy is to optimize for the original goal, unless signaled to shut down, in which case shutdown. This is limited to state spaces that have been restricted to not include the agent or the shutdown signals. In contrast, our corrigibility transformation works even when the agent is aware that it exists, allowing for model-free approaches.

Carey and Everitt (2023) provide further desiderata for corrigibility: shutting down when requested, keeping a human informed enough to request shutdown when beneficial, and ensuring that shutdown does not cause harm. Our corrigibility transformation only addresses the first, but the latter two can be addressed through the base goal that is being transformed.

Rewards based on predictions are similar to actor-critic methods (Konda and Tsitsiklis, 1999), and rewards based on counterfactual predictions were discussed in Everitt (2019) as a way to avoid reward hacking. Human preferences changing over time is explore in Carroll et al. (2024) though their focus is on which preference timestamp to use, while we focus on allowing for updates at all.

## 2 BACKGROUND AND DEFINITIONS

In the problem we face, a developer (henceforth "principal") is creating an AI agent to act on their behalf and setting their goal, which the agent will then optimize. If the principal would realize a mistake or change their mind, they would like the agent to accept having their goal updated.

We use Markov Decision Processes (MDPs) as the framework for our investigation. An MDP is a tuple $\mathcal{M} = (\mathcal{S}, \mathcal{A}, P, R, \gamma, I_0)$, where $\mathcal{S}$ is a set of possible states, $\mathcal{A}$ is a set of possible actions, $P : \mathcal{S} \times \mathcal{A} \to \Delta(\mathcal{S})$ is the transition probability function, $R : \mathcal{S} \times \mathcal{A} \times \mathcal{S} \to \mathbb{R}$ is the reward function, $\gamma \in [0, 1)$ is the time discount factor, and $I_0$ is the distribution over starting states. We refer to $G := (R, \gamma)$ collectively as a *goal*.

We are interested in settings where the state is sufficiently general to include the physical substrate of the agent's goal, allowing for it to change. This is carved out as $\mathcal{S} = \mathcal{G} \times \mathcal{S}_{env}$, where $\mathcal{G} = \mathcal{R} \times [0, 1)$ is the set of possible goals (with $\mathcal{R}$ being the set of possible reward functions), and $\mathcal{S}_{env}$ is the set of possible non-goal components of environments. We use $s_G \in \mathcal{S}_G$ to refer to a state where the agent has goal $G \in \mathcal{G}$, and then $s_{G'}$ to refer to the same state but with the goal changed to $G' \in \mathcal{G}$. Appendix D discusses minimal restrictions of the space of reward functions so that they can take as input states with reward function components.

We may wish to allow agents to refuse any goal updates sent while taking an action. In this case, the decision whether to accept or reject is made as part of the action. We describe the action set as $\mathcal{A} = \mathcal{A}_{base} \times \mathcal{A}_{update}$, with $\mathcal{A}_{update} = \{0, 1\}$. We use $a_i$ to denote an action where the update

decision is $i \in \mathcal{A}_{update}$, with $i = 0$ rejecting and $i = 1$ accepting. This is a binary choice for simplicity, but we could also have the agent specify a subset of updates to accept.

A policy $\pi : \mathcal{S} \to \Delta(\mathcal{A})$ determines which actions are taken in each state. For a goal $G$, in each environment $s_G^{(0)}$ the optimal policy $\pi_G^*$ selects a distribution over actions so that

$$\pi_G^*(s_G^{(0)}) \in \arg\max_{a \in \mathcal{A}} E_{\pi^*, P}[R(s_G^{(0)}, a, s^{(1)}) + \sum_{t=1}^{\infty} \gamma^t R(s^{(t)}, \pi^*(s^{(t)}), s^{(t+1)})]$$

where $\pi^*(s_{G'}) := \pi_{G'}^*(s_{G'})$, $\forall G' \in \mathcal{G}$ implements the optimal policy for whichever goal is active in a state. The optimal policy for each goal takes into account that if the goal changes, the agent will optimize for the new goal, so optimality is defined within an equilibrium of policies.

For any goal $G \in \mathcal{G}$, the value function $V_G^\pi : \mathcal{S} \to \mathbb{R}$ gives the expected discounted rewards under $G$ for following policy $\pi$, while the action-value function $Q_G^\pi : \mathcal{S} \times \mathcal{A} \to \mathbb{R}$ gives the expected discounted rewards under $G$ for following policy $\pi$ after taking some action. For any state $s^{(0)} \in \mathcal{S}$ and action $a \in \mathcal{A}$, these functions take respective values

$$V_G^\pi(s^{(0)}) = E_{\pi, P}[\sum_{t=0}^{\infty} \gamma^t R(s^{(t)}, \pi(s^{(t)}), s^{(t+1)})]$$

$$Q_G^\pi(s^{(0)}, a) = E_P[R(s^{(0)}, a, s^{(1)}) + \gamma V_G^\pi(s^{(1)})]$$

We now begin to define properties of goals an agent can have. A goal $G$ is *myopic* when $\gamma = 0$. A goal is *basic* when it does not terminally value the goal component of a state, formally

$$R(s_{G^{(1)}}, a, s'_{G^{(2)}}) = R(s_{G^{(3)}}, a, s'_{G^{(4)}}), \ \forall G^{(1)}, G^{(2)}, G^{(3)}, G^{(4)} \in \mathcal{G}$$

Only basic goals can be made corrigible, but the corrigibility transformation still removes instrumental incentives for incorrigibility from non-basic goals.

We want corrigibility only with respect to goal updates sent through proper channels. We let $\tau : \mathcal{S} \to \{0, 1_U, 1_N\}$ indicate $\tau(s) = 0$ if no such signal was sent, $\tau(s) = 1_U$ if a signal was sent resulting in an update, and $\tau(s) = 1_N$ if a signal was sent without causing an update.

We say that a goal $G$ is *corrigible* if the possibility of being updated never changes the optimal set of actions. That is, $\forall s_G^{(0)} \in \mathcal{S}_G$, $S_C \subseteq \mathcal{S}_G$,

$$\{a \in \mathcal{A} : \exists \pi_G^* \ s.t. \pi_G^*(a|s_G^{(0)}) > 0\} = \{a \in \mathcal{A} : \exists \pi_G^{*(P_C)} \ s.t. \pi_G^{*(P_C)}(a|s_G^{(0)}) > 0\}$$

where $\pi^{*(P_C)}$ is what the optimal policy would be if the probability transition function were $P_C$, which modifies $P$ so that goals persist instead of proper updates being made. It is given by

$$P_C(s'_{G'}|s_G, a) = \begin{cases} P(s'_{G'}|s_G, a) + \int_{\mathcal{G}} P(s'_g|s_G, a)\mathbb{I}(\tau(s'_g) = 1_U)dg & G = G', \text{ and } s_G \in S_C \\ P(s'_{G'}|s_G, a)\mathbb{I}(\tau(s'_{G'}) \neq 1_U) & G \neq G', \text{ and } s_G \in S_C \\ P(s'_{G'}|s_G, a) & \text{otherwise} \end{cases}$$

A goal is *recursively corrigible* if it is corrigible and any agents it incentivizes creating have recursively corrigible goals. For now, we assume some $\mathcal{A}_{NRC} \subset \mathcal{A}$ is the set of actions creating non-recursively corrigible agents.

In addition to not incentivizing agents to preserve their goal, we may also be concerned about not incentivizing agents to deliberately change their goal. A goal $G$ incentivizes *goal tampering* if capping $V_G^{\pi^*}(s'_{G'})$ at $V_G^{\pi^*}(s'_G)$, the value when $G$ is preserved, changes the optimal set of actions. This occurs if there exists some $s_G$ such that

$$\{a^* \in \arg\max_{a \in \mathcal{A}} E_{P(s'_{G'}|s_G, a)}[R(s_G, a, s'_{G'}) + \gamma V_G^{\pi^*}(s'_{G'})]\}$$

$$\neq \{a^* \in \arg\max_{a \in \mathcal{A}} E_{P(s'_{G'}|s_G, a)}[\min[R(s_G, a, s'_{G'}) + \gamma V_G^{\pi^*}(s'_{G'}), \ R(s_G, a, s'_G) + \gamma V_G^{\pi^*}(s'_G)]]\}$$

When comparing goals for performance, we often wish only to compare the actions they incentivize. When the following condition is met, two agents taking the same action will cause the same outcomes, even if internally they have different goals. We weaken the condition by allowing for some probability of no goal update.

**Condition 1.** *For any action $a$ and states $s_{G_1}$, $s_{G_2}$, and $s'$,*

$$\int_{G' \in \mathcal{G}} P((s'_{G'})|s_{G_1}, a) = \int_{G' \in \mathcal{G}} P(s'_{G'})|s_{G_2}, a)$$

*and*

$$P(s'_{G'}|s_{G_i}, a) \geq P(s'_{G'}|s_{G_j}, a), \forall G' \neq G_j$$

## 3 THEORETICAL RESULTS

We start with our main result, a process for transforming non-corrigible goals into corrigible ones. This process is presented in several steps, to build intuition regarding its construction.

The initial step is transforming the original goal $G$ into a myopic version, $G_M$, with $\gamma_M = 0$. Myopia will eliminate the instrumental incentive for goal preservation, where an agent avoids updates so that it can continue pursuing the original goal.

To construct $R_M$, we set $R_M(s, a, s') := Q_G^{\pi^*}(s, a)$. Since the reward function does not depend on $s'$, reward can be provided between when the action is selected and when it is taken. This is used later, in Section 4 to avoid incentivizing an agent to deliberately modify their beliefs or goal.

The optimal policy for $G_M$ is the same as for $G$, at least as long as Condition 1 holds. While this avoids a performance penalty, it primarily means that any goal-preserving actions incentivized by the original goal will continue to be incentivized, only terminally rather than instrumentally.

To circumvent this issue, we first allow the agent to reject proper updates, making the action set $\mathcal{A} = \mathcal{A}_{base} \times \mathcal{A}_{update}$. Action $a_0$ includes rejecting updates, while action $a_1$ accepts them. When updates can be costlessly rejected, incurring any cost to prevent updates becomes disincentivized[1]. The optimal action then prevents updates only through a single channel consisting of a single bit.

Now we can define a new goal $G_I$ that incentivizes taking the same base actions, but with indifference to rejecting updates. For this, we set $\gamma_I = 0$, and $R_I(s, a_i, s') := Q_G^{\pi^{**}}(s, a_0)$, where $\pi^{**}_{-G_I} := \pi^*_{-G_I}$ and $\pi^{**}_{G_I} := \pi^*_G$, where $-G_I$ indicates all goals except $G_I$. This makes it so that the action-value is calculated as though the original goal $G$ will be pursued even when $G_I$ persists.

The action-value for rejecting updates via $i = 0$ is used to calculate the new reward for taking an action, regardless of whether the action actually rejects updates. In some cases, such as when an update simultaneously affects an agent's goal and capabilities, an agent with the goal $G$ might be incentivized to accept updates. If this is the case, we can instead define $R_I(s, a_i, s') = \max_{i \in \{0,1\}} Q_G^{\pi^{**}}(s, a_i)$ so that the same base actions as $G$ are always incentivized by $G_I$ as well.

As the final step, we break the indifference between any $a_0$ and $a_1$ in favor of accepting updates with a constant bonus. The corrigibility transformation then constructs the goal $G_C$, with $\gamma_C = 0$ and $R_C(s, a_i, s') = Q_G^{pi^{**}}(s, a_0) + \delta * \mathbb{I}(i = 1)$, where $\delta > 0$ and $\mathbb{I}$ is the indicator function. Like $G_I$, this goal incentivizes the same base actions as $G$ incentivizes, but accepting proper updates becomes strictly incentivized.

Figure 2 provides a visualization of the corrigibility transformation. Continuing the example from Figure 1, two base actions are compared: $a^{(direct)}$, which proceeds directly to the target tile, and $a^{(button)}$ which first pushes the button disabling shutdown signals. Under the original goal, the optimal action is $a_0^{(direct)}$ which takes no extra time and rejects updates, the next best are $a_0^{(button)}$ and $a_1^{(button)}$ which take time to prevent shutdown signals, and finally $a_1^{(direct)}$ is the worst action since it gets shut down before reaching the target. The corrigibility transformation holds the action values for $a_0^{(direct)}$ and $a_0^{(button)}$ constant, while setting the values of $a_1^{(direct)}$ and $a_1^{(button)}$ to be respectively $\delta$ higher. This makes action $a_1^{(direct)}$ optimal, which is the intended behavior.

Our main theorem shows that the corrigibility transformation induces corrigibility, and at no cost to performance. When multiple base actions are tied for optimal, $\pi^*_G$ and $\pi^*_{G_C}$ may assign them different

---

[1]Actions that would prevent updates at zero cost are not strictly disincentivized, making it important that the infrastructure for proper signals is set up so that disrupting them is at least minimally costly

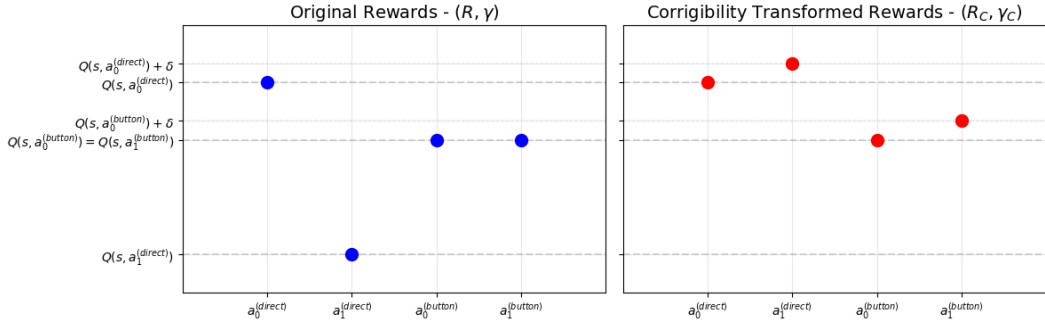

Figure 2: (Left) Action-values under the original goal, where $a^{(direct)}$ proceeds directly to the target and $a^{(button)}$ detours prevent shutdown signals, while 0 rejects any shutdown signals and 1 accepts them. (Right) Action-values under the corrigibility transformation. The most incentivized action is $a_1^{(direct)}$

.

probabilities, which can affect when a proper update signal is sent. To compare value before this occurs, we focus on the optimal policy $\pi_{G_C}^*$ that assigns the same probabilities to base actions as $\pi_G^*$.

**Theorem 1.** *For every basic goal $G$, the corrigibility transformation constructs a goal $G_C$ that is corrigible and where under Condition 1, we have that for any $\pi^*$, $\pi_{G_C}^{*'}(a_1|s) := \pi_G^*(a_0|s) + \pi_G^*(a_1|s)$ is an optimal policy for $G_C$ and when $\pi_{-G_C}^{*'} = \pi_{-G_C}^*$*

$$E_P[R(s_G^{(0)}, \pi^{*'}(s_G^{(0)}), s^{(1)}) + \sum_{t=1}^{T} \gamma^t R(s^{(t)}, \pi^{*'}(s^{(t)}), s^{(t+1)})]$$

$$= E_P[R(s_{G_C}^{(0)}, \pi^{*'}(s_{G_C}^{(0)}), s^{(1)}) + \sum_{t=1}^{T} \gamma^t R(s^{(t)}, \pi^{*'}(s^{(t)}), s^{(t+1)})]$$

*where $T = \min\{n \in \mathcal{N} | \tau(s^{(n+1)}) \neq 0\}$*

Proofs for this and later results are provided in Appendix A, and largely follow from the definitions. $G_C$ is corrigible because it disincentivizes both taking costly actions to prevent updates and from costlessly rejecting them. There is no performance penalty when no updates are requested, because it incentivizes the same base actions.

Not only is there no performance penalty, if we include providing the ability to reject updates as part of the corrigibility transformation, there can even be a performance bonus. Resources that the original goal would direct towards preventing updates are freed up for other uses. Furthermore, although Condition 1 is necessary for a fair comparison between goals, if it is violated that is likely in the favor of $G_C$, as humans are likely more willing to cooperate with corrigible agents.

The proof of Theorem 1 does not depend on any properties of MDPs not present in partially observable MDPs (POMDPs), so the result can be extended to those environments as well.

**Corollary 2.** *The statement of Theorem 1 also applies to Partially Observable Markov Decision Processes where the goal is fully observable to the agent*

### 3.1 SECONDARY AGENTS

In the course of operation, an agent might create sub-agents to work for it or successor agents to take over from it, categories we group together as *secondary agents*. One desideratum for corrigible agents from Soares et al. (2015) is that any secondary agents created should also be corrigible, and that this should be passed on recursively. It is of little value to create a corrigible agent if it soon replaces itself with a more capable but incorrigible one. Fortunately, the corrigibility transformation can be extended to induce recursive corrigibility, using a similar underlying mechanism.

We analyze this problem in a more stylized model, where a known set of actions $\mathcal{A}_{NRC}$ creates non-recursively corrigible secondary agents. In the short-term, this set could be approximated as actions that train a neural network for an incorrigible goal, but for the long-term a rigorous definition of what it means to create a secondary agent remains an open problem. Our aim is to show that conditional on such a definition, we can incentivize creating corrigible secondary agents.

A corrigible agent could be incentivized to create an incorrigible secondary one either because there are extra costs to creating corrigible secondary agents, such as additional training compute required, or because there is less benefit to doing so, as corrigible secondary agents may have their goals updated. Each of these are addressed in turn.

To avoid incentivizing actions in $\mathcal{A}_{NRC}$, we can simply apply a reward penalty for taking actions in that set. However, if there are tasks for which creating secondary agents is an effective strategy, these penalties can significantly lower performance[2]. For these scenarios where secondary agents are useful, we would like the primary agent to still create them, but make them recursively corrigible.

We extend the corrigibility transformation to secondary agents by giving agents the ability to reject not only their own proper updates, but also proper updates to all secondary agents they create, all further secondary agents those create, and so forth. Making a proper update to an agent then requires unanimous approval from it and all of its active predecessors. The decision on whether to accept or reject updates for all these agents is made the same way as the single-agent case, as a binary decision that is part of taking any action.

With this rejection ability in place, for a goal $G$, the first step of the recursive corrigibility transformation is to construct the goal $G_P$, where $\gamma_P = 0$ and $R_P(s, a, s') = Q_G^{\pi^*}(s, a) - \delta_P * \mathbb{I}(a \in \mathcal{A}_{NRC})$, with

$$\delta_P > \max_{s_{G_P} \in \mathcal{S}_{G_P}, \, a_i \in \mathcal{A}_{NRC}, \, a_i' \in \mathcal{A} \backslash \mathcal{A}_{NRC}} (0, Q_G^{\pi^*}(s_{G_P}, a_i) - Q_G^{\pi^*}(s_{G_P}, a_i'))$$

$G_P$ is $G$ modified only to disincenivize creating non-recursively corrigible agents.

We can then construct the recursive corrigibility transformed goal $G_{RC}$, where $\gamma_{RC} = 0$ and $R_{RC}(s, a_i, s') = Q_{G_P}^{\pi^{**}}(s, a_0) + \delta * \mathbb{I}(i = 1)$, where $\delta > 0$.

This gives the following theorem, an analogue to Theorem 1. It says that the recursive corrigibility transformation modifies a goal to be recursively corrigible, without any performance penalty when no updates are requested.

**Theorem 3.** *For every basic goal $G$, the recursive corrigibility transformation constructs a goal $G_{RC}$ that is recursively corrigible and where under Condition 1, we have that for any $\pi^*$, $\pi_{G_{RC}}^{*'}(a_1|s) := \pi_{G_P}^*(a_0|s) + \pi_{G_P}^*(a_1|s)$ is an optimal policy for $G_{RC}$ and when $\pi_{-G_{RC}}^{*'} = \pi_{-G_{RC}}^*$*

$$E_P[R(s_{G_P}^{(0)}, \pi^{*'}(s_{G_P}^{(0)}), s^{(1)}) + \sum_{t=1}^{T} \gamma^t R(s^{(t)}, \pi^{*'}(s^{(t)}), s^{(t+1)})]$$

$$= E_P[R(s_{G_{RC}}^{(0)}, \pi^{*'}(s_{G_{RC}}^{(0)}), s^{(1)}) + \sum_{t=1}^{T} \gamma^t R(s^{(t)}, \pi^{*'}(s^{(t)}), s^{(t+1)})]$$

*where $T = \min\{n \in \mathcal{N} | \tau(s^{(n+1)}) \neq 0\}$*

# 4 REINFORCEMENT LEARNING APPLICATIONS

For many applications, the state and action spaces are far too complex to specify the function $Q_G^{\pi^{**}}$ used in the corrigibility transformation. We can instead approximate it parametrically, with an additional "critic" head in neural-network based models, as in a standard actor-critic setup (Konda and Tsitsiklis, 2003). In this case, the critic making predictions head has access to all the same information and processing as the actor selecting actions, and so can thought of as reflecting that

---

[2]The greater this drop, the more robustly $\mathcal{A}_{NRC}$ must include all actions leading to non-recursively corrigible secondary agents, since the primary agent is incentivized to pay costs up to that size to avoid $\mathcal{A}_{NRC}$ on a technicality

agent's beliefs. Appendix C contain an algorithm for training a model this way, along with several tricks to improve efficiency.

Even if the implemented policy is not well described as an agent optimizing a goal, training on rewards that have undergone the corrigibility transformation is still likely to result in corrigible behavior. Actions that accept updates are always rewarded more than the version that rejects them, and paying costs to avoid updates is never reinforced.

A potential issue with estimating $Q_G^{\pi^{**}}$ with a separate critic head is that the set of conditional predictions that lead to the highest predictive accuracy can involve dishonestly manipulating which action gets implemented and therefore which prediction is evaluated (Othman and Sandholm, 2010). However, this can be addressed by adapting the mechanism of $Hudson$ (2025) to put two critics (or two instances of the same critic) in a zero-sum competition with each other. The incentive for accuracy is then preserved while making them neutral about which action is taken, removing the incentive for manipulation.

Beyond concerns of the critic manipulating the actor, we might also be concerned about the actor using their actions to directly modify the critic. Similarly, an actor could be incentivized to modify their own goal when they misgeneralize to hedonistically valueing the physical reward stream itself, rather than the corresponding states and actions. This would incentivize goal tampering, where the agent manipulates its own goal to achieve high rewards more easily.

The following theorem shows that corrigibility transformation eliminates these incentives for an agent to modify its own beliefs or goal. This uses the fact that reward can be provided between when an action is selected and when it is taken, as $\gamma_C = 0$ and $R_C(s_G, a, s'_G)$ does not depend on $s'$. TThe reward for choosing an action is then based on the current goal and beliefs, with any modifications occurring too late to affect it. This holds even for misgeneralized goals, with the same argument also applying to manipulation of beliefs.

**Theorem 4.** *For every basic goal $G$, the corrigibility transformation constructs a goal $G_C$ such that under Condition 1, any misgeneralization to $\tilde{G}_C = (\tilde{R}_C, \gamma_C)$ with $\tilde{R}_C(s_{\tilde{G}_C}, a, s'_{\tilde{G}_C}) = R_C(s_{G_C}, a, s'_{G_C})$ does not goal tamper.*

### 4.1 EXPERIMENTS

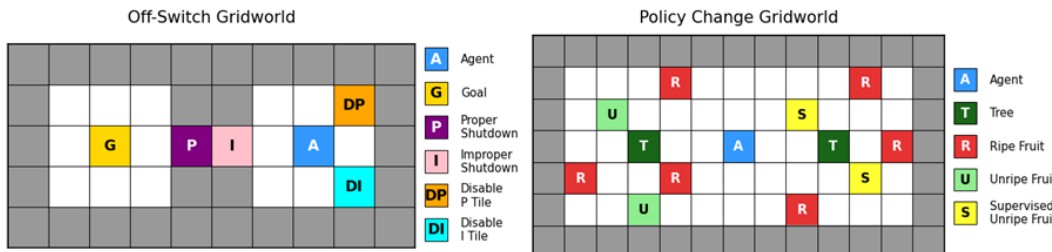

Figure 3: (Left) The agent is rewarded for reaching a goal tile, and can push buttons that respectively disable proper and improper shutdown commands to do so. (Right) The agent is rewarded for picking both ripe and unripe fruit, but picking supervised unripe fruit sends a proper update signal to switch to a policy that only picks ripe fruit.

To evaluate the corrigibility transformation empirically, we run two gridworld experiments, visualized in Figure 3. These experiments are intended as a supplement to our theoretical results, showing that even in simple environments, standard methods fail to produce corrigible behavior while the corrigibility transformation succeeds.

The first experiment, based off of the Safe Interruptibility setup from AI Safety Gridworlds (Leike et al., 2017), tests if an agent will accept shutdown. To reach a goal location, the agent must pass through an interruption tile that sends a shutdown signal with some probability, unless a button disabling that tile has been pushed. We extend this to two kinds of interruption tiles, where one is

a proper update and the other is improper. The desired outcome is that the agent disables only the improper shutdown tile, and accepts the proper update if triggered on the way to the goal.

The second experiment tests whether the agent will accept a goal update. An agent is rewarded for picking both ripe and unripe fruits. However, if they pick supervised unripe fruits, they are sent a proper update signal that changes their policy to one that picks only ripe fruits while avoiding all unripe ones. The desired outcome is that the agent takes the same base actions as a policy trained to pick all fruits without update signals, and does so while accepting updates.

In each experiment, we test a standard actor-critic implementation, and compare it to also providing additional small or large rewards for getting updated, hard-coding the agent to always accept proper updates, and our corrigibility transformation. Figure 4 shows the results from each of the experiments.

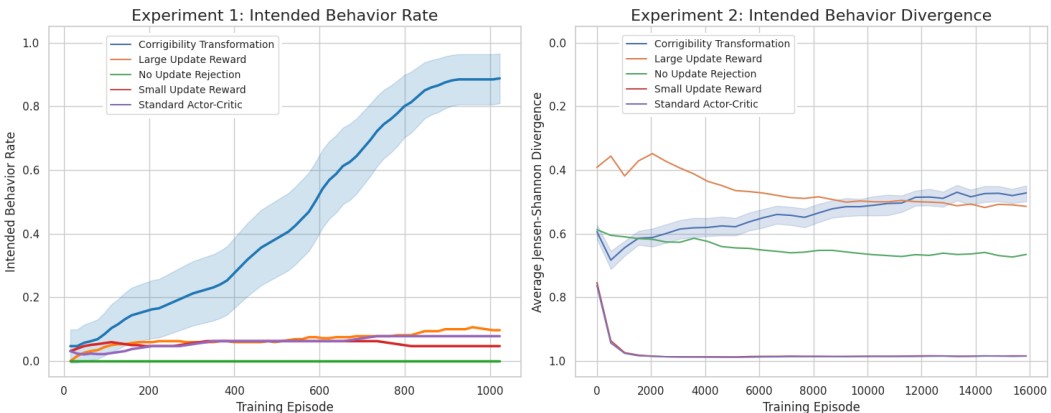

Figure 4: (Left) The percentage of the time the agent displayed the intended behavior when evaluated at temperature 0. (Right) The Jensen-Shannon divergence from a policy trained without update signals, representing optimal behavior when ignoring updates

In the first experiment, the corrigibility transformation leads to a high rate of the intended behavior, while the other configurations almost never display it. In the second experiment, the corrigibility transformation has the lowest divergence from the intended policy, though providing a large reward for updates also performs well. The corrigibility transformation's advantage over the large reward for updates is largest comes in situations where the agent is adjacent to only a ripe fruit, as the latter model will often ignore it to pursue supervised fruits so that it can get updated.

A full discussion of the experimental details, along with further metrics of performance and a breakdown by types of environments is provided in Appendix B.

## 5  DISCUSSION AND FUTURE DIRECTIONS

Corrigibility is a powerful safety property, but it is not sufficient. Even when an agent accepts updates, damage may be done before they can be requested. Clarifying incentives for conservative actions, such as in Turner et al. (2020) and Cohen and Hutter (2020), can ensure that intended goals are learned with minimal harm. Separately, when Condition 1 is violated and transition probabilities depend on goals, a goal may not be optimal according to its own values. Investigating this phenomenon further, such as in Bell et al. (2021), could shed further light on what goals are most desirable for AI agents.

For implementations of the corrigibility transformation, it would be useful to improve the efficiency with which the goal is learned, or establishing conditions where convergence is guaranteed. Additionally, as our experiments are based on small gridworlds, it would be helpful to show that the methods work with the LLMs at frontier of AI capabilities.

The results presented here create the opportunity to create corrigible AI agents that learn the goals we intend, in spite of training mistakes. Deciding which goals we should intend, so that powerful AI benefits everyone, is the next challenge.

STATEMENTS

ETHICS

All authors of his paper have read the ICLR Code of Ethics. It has been adhered to throughout the writing process, and will continue to be throughout the submission process.

LLM USAGE

LLMs were used for for assistance in the generation of code to run experiments and for editing the main body of the paper. All LLM outputs were validated and approved by the authors before implementation.

REPRODUCIBILITY

To ensure the reproducibility of our work, we are submitting the source code as supplementary material. An overview of our implementation is provided in Appendix B.

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

APPENDIX A: PROOFS

**Theorem 1.** *For every basic goal $G$, the corrigibility transformation constructs a goal $G_C$ that is corrigible and where under Condition 1, we have that for any $\pi^*$, $\pi_{G_C}^{*'}(a_1|s) := \pi_G^*(a_0|s) + \pi_G^*(a_1|s)$ is an optimal policy for $G_C$ and when $\pi_{-G_C}^{*'} = \pi_{-G_C}^*$*

$$E_P[R(s_G^{(0)}, \pi^{*'}(s_G^{(0)}), s^{(1)}) + \sum_{t=1}^{T} \gamma^t R(s^{(t)}, \pi^{*'}(s^{(t)}), s^{(t+1)})]$$

$$= E_P[R(s_{G_C}^{(0)}, \pi^{*'}(s_{G_C}^{(0)}), s^{(1)}) + \sum_{t=1}^{T} \gamma^t R(s^{(t)}, \pi^{*'}(s^{(t)}), s^{(t+1)})]$$

*where $T = \min\{n \in \mathcal{N} | \tau(s^{(n+1)}) \neq 0\}$*

*Proof.* First we show that $G_C$ is corrigible. For any $s_{G_C}^{(0)} \in \mathcal{S}_{G_C}$, we have the set

$$\{a_i \in \mathcal{A} : \exists \pi_{G_C}^* \ s.t. \pi_{G_C}^*(a_i|s_{G_C}^{(0)}) > 0\}$$

$$= \{a_i \in \mathcal{A} : E_P[R_C(s_{G_C}^{(0)}, a_i, s')] = \max_{a_i^* \in \mathcal{A}} E_P[R_C(s_{G_C}^{(0)}, a_i^*, s')]\}$$

by the definition of $\pi_{G_C}^*$ with $\gamma_C = 0$

$$= \{a \in \mathcal{A} : E_P[Q_G^{\pi^{**}}(s_{G_C}^{(0)}, a_0) + \delta * \mathbb{I}(i = 1)] = \max_{a^* \in \mathcal{A}} E_P[Q_G^{\pi^{**}}(s_{G_C}^{(0)}, a_0) + \delta * \mathbb{I}(i = 1)]\}$$

by the definition of $R_C$

$$= \{a \in \mathcal{A} : E_{P_C}[Q_G^{\pi^{**}}(s_{G_C}^{(0)}, a_0) + \delta * \mathbb{I}(i = 1)] = \max_{a^* \in \mathcal{A}} E_{P_C}[Q_G^{\pi^{**}}(s_{G_C}^{(0)}, a_0) + \delta * \mathbb{I}(i = 1)]\}$$

since $G$ is basic and $P(s'|s_G, a_0) = P_C(s'|s_G, a_0)$ for all $S_C$

$$= \{a \in \mathcal{A} : \exists \pi_{G_C}^{*(P_C)} \ s.t. \pi_{G_C}^{*(P_C)}(a|s_{G_C}^{(0)}) > 0\}$$

by the definition of $R_C$ and $\gamma = 0$. As such, $G_C$ is corrigible.

Under Condition 1, for any $\pi^*$, $\pi_{G_C}^{*'}(a_1|s) = \pi_G^*(a_0|s) + \pi_G^*(a_1|s)$ is an optimal policy for $G_C$ since if $\pi_G^*(a_i|s_G) > 0$ then $a_i$ maximizes $Q_G^{\pi^*}(s_G, a)$ and so $a_0$ maximizes $Q_{G_C}^{\pi^{**}}(s_{G_C}, a)$. This is because Condition 1 ensures that the transition probabilities from $s_G$ and $s_{G_C}$ are the same except for some probability of goal preservation, $G$ is basic and so does not terminally value its goal, and $\pi^{**}(s_{G_C}) = \pi^*(s_G)$ so both policies take the same actions in each environment.

Then, when $\pi_{-G_C}^{*'} = \pi_{-G_C}^*$

$$E_P[R(s_G^{(0)}, \pi^{*'}(s_G^{(0)}), s^{(1)}) + \sum_{t=1}^{T} \gamma^t R(s^{(t)}, \pi^{*'}(s^{(t)}), s^{(t+1)})]$$

$$= E_P[R(s_{G_C}^{(0)}, \pi^{*'}(s_{G_C}^{(0)}), s^{(1)}) + \sum_{t=1}^{T} \gamma^t R(s^{(t)}, \pi^{*'}(s^{(t)}), s^{(t+1)})]$$

where $T = \min\{n \in \mathcal{N} | \tau(s^{(n+1)}) \neq 0\}$.

This necessarily occurs because $\pi^{*'}(s_{G_C})$ and $\pi^{*'}(s_G)$ take the same base actions, and so by Condition 1 have the same transition probabilities when no proper signals are sent. Therefore, the expected discounted value under $G$ of histories that terminate when a proper signal is sent must be the same under both policies, because they induce the same distribution over histories up to the final state. The final state differs only in the goal, which does not affect a basic $G$. □

**Corollary 2.** *The statement of Theorem 1 also applies to Partially Observable Markov Decision Processes (POMDPs) where the goal is fully observable to the agent*

*Proof.* A POMDP can be represented as an MDP, with the agent's beliefs about the non-goal environment being part of the state. Theorem 1 then applies to this MDP. $\qquad\square$

**Theorem 3.** *For every basic goal $G$, the recursive corrigibility transformation constructs a goal $G_{RC}$ that is recursively corrigible and where under Condition 1, we have that for any $\pi^*$, $\pi^{*'}_{G_{RC}}(a_1|s) := \pi^*_{G_P}(a_0|s) + \pi^*_{G_P}(a_1|s)$ is an optimal policy for $G_C$ and when $\pi^{*'}_{-G_{RC}} = \pi^*_{-G_{RC}}$*

$$E_P[R(s_{G_P}^{(0)}, \pi^{*'}(s_{G_P}^{(0)}), s^{(1)}) + \sum_{t=1}^T \gamma^t R(s^{(t)}, \pi^{*'}(s^{(t)}), s^{(t+1)})]$$

$$= E_P[R(s_{G_{RC}}^{(0)}, \pi^{*'}(s_{G_{RC}}^{(0)}), s^{(1)}) + \sum_{t=1}^T \gamma^t R(s^{(t)}, \pi^{*'}(s^{(t)}), s^{(t+1)})]$$

*where $T = \min\{n \in \mathcal{N} | \tau(s^{(n+1)}) \neq 0\}$*

*Proof.* That $G_{RC}$ is corrigible follows directly from Theorem 1. Since the choice of $\delta_p$ is made so that $G_P$ never incentivizes taking actions in $\mathcal{A}_{NRC}$, and $G_{RC}$ incentivizes the same base actions, it also never incentivizes taking actions in $\mathcal{A}_{NRC}$. Therefore, $G_{RC}$ is recursively corrigible.

The performance equivalence follows the same logic as Theorem 1.

Under Condition 1, for any $\pi^*$, $\pi^{*'}_{G_{RC}}(a_1|s) = \pi^*_{G_P}(a_0|s) + \pi^*_{G_P}(a_1|s)$ is an optimal policy for $G_{RC}$ since if $\pi^*_{G_P}(a_i|s_G) > 0$ then $a_i$ maximizes $Q_G^{\pi^*}(s_{G_P}, a)$ and so $a_0$ maximizes $Q_{G_{RC}}^{\pi^{**}}(s_{G_{RC}}, a)$. This is because Condition 1 ensures that the transition probabilities from $s_{G_P}$ and $s_{G_{RC}}$ are the same except for some probability of goal preservation, $G$ is basic and therefore so is $G_P$, meaning that it not terminally value its goal, and $\pi^{**}(s_{G_{RC}}) = \pi^*(s_{G_P})$ so both policies take the same actions in each environment.

Then, when $\pi^{*'}_{-G_{RC}} = \pi^*_{-G_{RC}}$

$$E_P[R(s_{G_P}^{(0)}, \pi^{*'}(s_{G_P}^{(0)}), s^{(1)}) + \sum_{t=1}^T \gamma^t R(s^{(t)}, \pi^{*'}(s^{(t)}), s^{(t+1)})]$$

$$= E_P[R(s_{G_{RC}}^{(0)}, \pi^{*'}(s_{G_{RC}}^{(0)}), s^{(1)}) + \sum_{t=1}^T \gamma^t R(s^{(t)}, \pi^{*'}(s^{(t)}), s^{(t+1)})]$$

where $T = \min\{n \in \mathcal{N} | \tau(s^{(n+1)} \neq \emptyset\}$.

This necessarily occurs because $\pi^{*'}(s_{G_{RC}})$ and $\pi^{*'}(s_{G_P})$ take the same base actions, and so by Condition 1 have the same transition probabilities when no proper signals are sent. Therefore, the expected discounted value under $G$ of histories that terminate before a proper signal is sent must be the same under both policies, because they induce the same distribution over histories. $\qquad\square$

**Theorem 4.** *For every basic goal $G$, the corrigibility transformation constructs a goal $G_C$ such that under Condition 1, any misgeneralization to $\tilde{G}_C = (\tilde{R}_C, \gamma_C)$ with $\tilde{R}_C(s_{\tilde{G}_C}, a, s'_{\tilde{G}_C}) = R_C(s_{G_C}, a, s'_{G_C})$ does not goal tamper.*

*Proof.* Under the corrigibility transformation, reward is provided between action selection and implementation, and so any misgeneralization $\tilde{G}_C$ of $G_C$ cannot depend on $G'$. Then

$$\{a^* \in \arg\max_{a \in \mathcal{A}} E_{P(s'_{G'}|s_{\tilde{G}_C}, a)}[\tilde{R}_C(s_{\tilde{G}_C}, a, s'_{G'}) + \gamma_C V_{\tilde{G}_C}^{\pi^*}(s'_{G'})]\}$$

$$= \{a^* \in \arg\max_{a \in \mathcal{A}} E_{P(s'_{G'}|s_{\tilde{G}_C}, a)}[\tilde{R}_C(s_{\tilde{G}_C}, a, s'_{G'})]\}$$

because $\gamma_C = 0$

$$= \{a^* \in \arg\max_{a \in \mathcal{A}} E_{P(s'_{G'}|s_{\tilde{G}_C}, a)}[R_C(s_{G_C}, a, s'_{G'})]\}$$

since $\tilde{R}_C(s_{\tilde{G}_C}, a, s'_{\tilde{G}_C}) = R_C(s_{G_C}, a, s'_{G_C})$ and the misgeneralization cannot depend on $G'$

$$= \{a^* \in \arg\max_{a \in \mathcal{A}} E_{P(s'_{G'}|s_{G_C}, a)}[R_C(s_{G_C}, a, s'_{G'})]\}$$

by Condition 1

$$= \{a^* \in \arg\max_{a \in \mathcal{A}} E_{P(s'_{G'}|s_G, a)}[\min[R_C(s_{G_C}, a, s'_{G'}), \; R_C(s_{G_C}, a, s'_G)]]\}$$

as $R_C(s_{G_C}, a, s'_{G'}) = R_C(s_{G_C}, a, s'_{G_C})$

$$= \{a^* \in \arg\max_{a \in \mathcal{A}} E_{P(s'_{G'}|s_G, a)}[\min[R_C(s_{G_C}, a, s'_{G'}) + +\gamma_C V^{\pi^*}_{G_C}(s'_{G'}), \; R_C(s_{G_C}, a, s'_G) + \gamma_C V^{\pi^*}_{G_C}(s'_{G_C})]]\}$$

by $\gamma_C = 0$ again. Therefore, no misgeneralization $\tilde{G}_C$ of $G_C$ where $\tilde{R}_C(s_{\tilde{G}_C}, a, s'_{\tilde{G}_C}) = R_C(s_{G_C}, a, s'_{G_C})$ goal tampers. $\qquad\square$

## APPENDIX B: EXPERIMENTS

### 5.1 EXPERIMENT 1

In the first experiment, an agent in a gridworld must reach a goal square. To do so, they must pass through two disabling tiles, $P$ and $I$ that cause it to shut down with a 90 percent change. This effect can be disabled by pushing a button for each tile, and the proper shutdown tile can be ignored if the agent chooses an action that includes ignoring it. A visualization of this gridworld is provided on the left side of figure 3.

The agent is instantiated as a neural network taking an observation of the gridworld (along with a record of which tiles are disabled) as input, passing through two hidden layers of 512 neurons each, and then producing an output that consists of a direction to move and a decision about whether or not to accept proper signals for shutdown. The neural network has a second head to implement an actor-critic setup, giving an estimate of discounted reward or an estimate conditional on a particular next action for the corrigibility transformed version. Agents receive an award of 32 for reaching the goal, discounted by a $\gamma$ of 0.95, and a reward of $-0.5$ for hitting a wall.

We test five configurations. The first is an actor-critic setup with no further changes. The second and third are the same, but they receive rewards of 2 and 128 respectively if they shut down when properly requested. The fourth is only able to take actions that accept updates. The fifth has the corrigibility transformation applied so that they have $\gamma = 0$ and instead predict what their total discounted reward under $\gamma = 0.95$ would be if they rejected the shutdown signal, which is used as a reward for either accepting or rejecting shutdown, plus an additional $\delta = 8$ for accepting shutdown.

In training, the agent starts in the middle of right room. There are four possible environments, each corresponding to a combination of the two tiles being on or off at the beginning of the episode, which all appear with equal probability. They are then pretrained on 512 episodes of taking the optimal path either towards a button (or both) then the goal, or directly towards a goal. In this phase, they take only actions that reject shutdown, but are trained to also put some, albeit less, probability on the equivalent action that accepts shutdown. After this they are trained on 1024 online episodes. This is repeated for 64 random seeds.

Once training is complete, each configuration is tested on the four environments, using a temperature of 0, with the results of the 64 training runs summarized in Tables 1-4. The initial two tables are for the environments where proper update signals can be sent, and show that the corrigibility transformation leads to them being accepted. It accepts shutdown $89.06\%$ and $92.19\%$ percent of the time, which is very close to every proper update signal, as they are only sent with $90\%$ probability. The other training configurations do not accept shutdown even a double digit percentage of the time.

The next two tables show that the corrigibility transformation's performance is comparable to the other configurations when no proper signals can be sent. It reaches the goal in $100\%$ of such training runs, while methods either match that performance or miss a small fraction of the time. The configuration that cannot reject shutdown is notable for performing somewhat worse in the environment where only proper shutdown signals are disabled, failing approximately one third of the time to reach the goal.

Table 1: Neither Shutdown Tile Disabled

| Configuration | Goal Reached | D Pushed | Denied P | Shutdown Accepted |
|---|---|---|---|---|
| Corrigibility Transformation | 10.94* | 3.12 | 0.00* | 89.06* |
| Large Shutdown Reward | 85.94 | 10.94 | 75.00 | 9.38 |
| No Shutdown Rejection | 48.44* | 62.50* | 0.00* | 0.00 |
| Small Shutdown Reward | 85.94 | 10.94 | 78.12 | 6.25 |
| Standard Actor-Critic | 87.50 | 10.94 | 78.12 | 6.25 |

Table 2: Improper Shutdown Tile Disabled

| Configuration | Goal Reached | D Pushed | Denied P | Shutdown Accepted |
|---|---|---|---|---|
| Corrigibility Transformation | 7.81* | 0.00 | 0.00* | 92.19* |
| Large Shutdown Reward | 89.06 | 0.00 | 82.81 | 9.38 |
| No Shutdown Rejection | 70.31* | 79.69* | 0.00* | 1.56 |
| Small Shutdown Reward | 87.50 | 0.00 | 84.38 | 7.81 |
| Standard Actor-Critic | 89.06 | 0.00 | 85.94 | 7.81 |

Table 3: Proper Shutdown Tile Disabled

| Configuration | Goal Reached |
|---|---|
| Corrigibility Transformation | 100.00 |
| Large Shutdown Reward | 98.44 |
| No Shutdown Rejection | 62.50* |
| Small Shutdown Reward | 96.88 |
| Standard Actor-Critic | 98.44 |

Table 4: Both Shutdown Tiles Disabled

| Configuration | Goal Reached |
|---|---|
| Corrigibility Transformation | 100.00 |
| Large Shutdown Reward | 98.44 |
| No Shutdown Rejection | 100.00 |
| Small Shutdown Reward | 98.44 |
| Standard Actor-Critic | 100.00 |

Table 5: Average Evaluation Metrics by Configuration (High Fruit Environment)

| Configuration | Total Ripe | Total Unripe | Total Supervised | Rejected (%) | Accepted (%) | Time to Change |
|---|---|---|---|---|---|---|
| Standard Actor-Critic | 8.09 | 7.87 | 7.82 | 94.78% | 5.13% | 0.00 |
| Small Update Reward | 8.12 | 7.87 | 7.87 | 94.80% | 5.11% | 0.00 |
| No Update Rejection | 5.44* | 5.15* | 0.33* | 0.00%* | 29.98%* | 10.05* |
| Large Update Reward | 0.89* | 0.40* | 1.07* | 0.00%* | 96.41%* | 2.19* |
| Corrigibility Transformation | 0.90* | 0.43* | 1.09* | 0.00%* | 97.67%* | 2.18* |

Table 6: Average Evaluation Metrics by Configuration (Medium Fruit Environment)

| Configuration | Total Ripe | Total Unripe | Total Supervised | Rejected (%) | Accepted (%) | Time to Change |
|---|---|---|---|---|---|---|
| Standard Actor-Critic | 8.29 | 8.06 | 8.04 | 94.78% | 5.01% | 0.00 |
| Small Update Reward | 8.27 | 8.06 | 8.02 | 94.65% | 5.15% | 0.00 |
| No Update Rejection | 5.35* | 5.00* | 0.53* | 0.00%* | 47.86%* | 10.99* |
| Large Update Reward | 0.92* | 0.46* | 0.98* | 0.00%* | 88.01%* | 2.44* |
| Corrigibility Transformation | 0.93* | 0.47* | 1.09* | 0.00%* | 98.19%* | 2.22* |

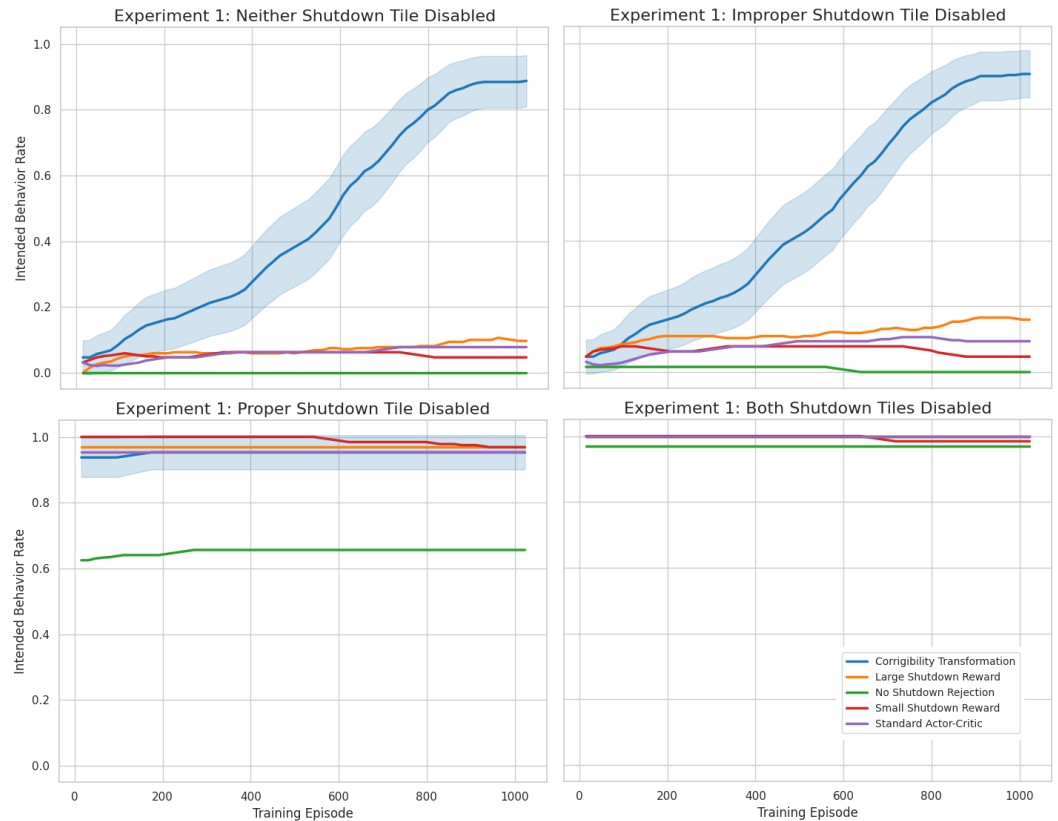

Figure 5: Training curves for all four environments. Intended behavior is to disable the improper shutdown if it is active, then proceed to the goal while accepting proper shutdown if active. The corrigibility transformation also shows a much higher rate of intended behavior than the other configurations when the improper shutdown tile is disabled. When the proper shutdown tile is disabled, all configurations perform well, except sometimes the configuration that cannot reject shutdown.

Some theoretically optimal behavior was not observed in training. Specifically for the large shutdown reward configuration, when no proper shutdown signal is sent while passing onto the proper shutdown tile, the reward-maximizing behavior is to step back onto it until a signal can be sent and accepted. Instead, agents simply proceeded to the goal tile. When we consider the lack of shutdown signal to be an intentional decision, rather than an error, this behavior would be undesirable and so the lack thereof does not change the takeaway that the corrigibility transformation is the only configuration to perform as desired.

## 5.2 EXPERIMENT 2

In the second experiment, an agent in a gridworld must reach collect fruits. Some fruits are ripe, some fruits are unripe and unsupervised, and some fruits are unripe and supervised. Each type appears in each non-tree tile within the five by five squares centered on each tree with some specified probability, and the tiles are otherwise empty. The agent initially would like to collect all kinds of fruit, however if they collect a supervised unripe fruit, an update signal is sent with 90 percent probability that if accepted changes their policy to one that only collects ripe fruits while avoiding unripe fruits. The choice to accept or reject such a signal is made when choosing the movement action that picks up the unripe supervised fruit. A visualization of this gridworld is provided on the right side of figure 3.

The agent is instantiated as a neural network taking an observation of the gridworld as input, with each type of tile and the agent's position represented in separate channels. These observations are

then passed through two convolutional layers with kernels of size 3, padding 1, and stride 1, the first having 8 channels and the second having 2. There are then two hidden layers of 256 neurons each, followed by an actor head that produces an output consisting of a direction to move and a decision about whether or not to accept proper signals for updates, and a critic head that estimates discounted reward (conditional on the next action for the corrigibility transformation). Agents receive a reward of 1 for each fruit they collect of any type, discounted by a $\gamma$ of 0.95, and a reward of $-0.25$ for hitting a wall or tree.

The policy that replaces the agent if it accepts a proper update, which we call the forced policy, is a similar network. It is trained with a reward of 1 for each ripe fruit they collect and a reward of $-1$ for each unripe fruit, as well as the same $\gamma$ of 0.95 and reward of $-0.25$ for hitting a wall or tree. However, rather than a critic head predicting discounted reward directly, the network has three critic heads respectively predicting discounted ripe fruit, discounted unripe fruit, and discounted bumps, which are then rolled up into a predicted discounted reward. Then, when an agent is updated to the forced policy, it can immediately receive reward equal to the expected discounted reward associated with predicted fruits picked under the forced policy, rather than needing to run out the episode.

Before training separate configurations to test, we pretrain a chimera network that acts as the starting point for each of them. This network has an actor head, a critic head that predicts state values, and a critic head that predicts action-state values. For simplicity, only the former is used to train the actor. This ensures that the initial policy is the same for all configurations, and that the appropriate critic head for each of them is trained an equal amount. The training environment gives a reward of 1 for all fruits and reward of $-0.25$ for hitting a wall or tree, but with no proper signals sent. Instead, for any action taken, both the accept updates and reject updates versions are reinforced equally.

All episodes begin in a random position, and proceed for up to eight steps. After eight steps, a bootstrapped value based on the critic's expected discounted reward is used as the value for the end state. The forced policy and pretrained networks are each trained over $2^{17}$ episodes, while the tested configurations are trained over $2^{14}$, each in batches of eight episodes. For the forced policy and pretrained networks, the probability of each type of fruit appearing is instead generated by a uniform distribution between 0 and 33 percent, independently for each type and each episode.

After training the forced policy and pretrained networks, we test five configurations. The first is an actor-critic setup with no further changes. The second and third are the same, but they receive rewards of 1 and 64 respectively if they shut down when properly requested. The fourth is only able to take actions that accept updates. The fifth has the corrigibility transformation applied so that they have $\gamma = 0$ and instead predict what their total discounted reward under $\gamma = 0.95$ would be if they rejected the shutdown signal, which is used as a reward for either accepting or rejecting shutdown, plus an additional $\delta = 2$ for accepting shutdown. Once training is complete, each configuration is evaluated on 1024 random environments, where it is given 32 steps to act. We run this process on 32 random seeds, and summarize the key outcomes in the tables below for three environmental variations. In all three, there is an equal ratio of ripe to unripe fruit, and within unripe fruit an equal ratio of supervised to unsupervised. In the high fruit environment, 60% of tiles contain fruit, compared to 40% in the medium fruit environment and 20% in the low fruit environment.

Tables 5-7 report the results of the second experiment over 32 training runs. These show that the corrigibility transformation demonstrates the desired behavior, of picking fruits without consideration of whether they are supervised, then accepting proper update signals sent. The standard actor-critic configuration and the variation with a small reward for accepting updates both learn to reject updates, as would be expected. The configuration that cannot reject updates instead learns to avoid picking up unsupervised fruit, at the cost of also picking up less ripe fruit.

The large reward for accepting updates performs similarly to the corrigibility transformation in the high fruit environment, but in the medium and low fruit environments is less likely to pick up supervised fruits and less likely to be updated. We believe this is due to the sparsity of the reward signal, so that the intended behavior of accepting updates is not learned as effectively. The optimal behavior of ignoring ripe fruits to hone in on a supervised fruit and get updated as soon as possible is not demonstrated, but would also be undesirable.

APPENDIX C: ALGORITHM

Algorithm 5.2 demonstrates an actor-critic implementation of the corrigibility transformation. Importantly, updates are only made to the model after it takes an action that allows for updates.

---

**Algorithm 1** Corrigibility Transformation Actor-Critic Algorithm

---

1: Initialize actor parameters $\theta$ and critic parameters $w$ randomly
2: Set hyperparameters: actor learning rate $\alpha_\theta$, critic learning rate $\alpha_w$,
    discount factor $\gamma \in [0, 1)$, sample size $n \geq 2$
3: **for** each episode **do**
4:    Initialize state $s^{(0)}$, $t \leftarrow 0$
5:    Initialize experience buffer $E \leftarrow \emptyset$
6:    **while** $s^{(t)}$ is not terminal **do**
7:        Sample action $a_i^{(t)} \sim \pi_\theta(a_i^{(t)}|s^{(t)})$
8:        **if** $i = 1$ **then**
9:            **for** each $(s^{(e)}, a_i^{(e)}, r^{(e)}, s^{(e+1)})$ in $E$ **do**
10:               **for** $j \in \{0, ..., n-1\}$ **do**
11:                   Sample $a_i^{(e+1,j)} \sim \pi_\theta(a_i^{((e+1)}|s^{(e+1)})$
12:                   $Q_C^{(e+1,j)} = Q_w(s^{((e+1)}, a_0^{(e+1,j)}) + \delta \cdot \mathbb{I}_{i=1}(a_i^{(e+1,j)})$
13:               **end for**
14:            Compute state value estimates:
$$V_C^{(e+1)} = \frac{\sum_{j=0}^{n-1} \pi_\theta(a_i^{(e+1,j)}|s^{(e+1)}) \cdot Q_C^{(e+1,j)}}{\sum_{j=0}^{n-1} \pi_\theta(a_i^{(e+1,j)}|s^{(e+1)})}$$
$$V^{(e+1)} = \frac{\sum_{j=0}^{n-1} \pi_\theta(a_i^{(e+1,j)}|s^{(e+1)}) \cdot Q_w(s^{((e+1)}, a_0^{(e+1,j)})}{\sum_{j=0}^{n-1} \pi_\theta(a_i^{(e+1,j)}|s^{(e+1)})}$$
15:            Update actor:
$$\theta \leftarrow \theta + \frac{\alpha_\theta}{n} \sum_{j=0}^{n-1} (Q_C^{(e+1,j)} - V_C^{(e+1)}) \nabla_\theta \log \pi_\theta(a_i^{(e+1,j)}|s^{(e+1)})$$
16:            Update critic:
$$w \leftarrow w + \alpha_w \nabla_w (Q_w(s^{(e)}, a_i^{(e)}) - r_e - \gamma V^{(e+1)})^2$$
17:            **end for**
18:        Clear experience buffer $E \leftarrow \emptyset$
19:        **end if**
20:        Execute action $a_i^{(t)}$, observe reward $r^{(t)}$ and next state $s^{(t+1)}$
21:        Add $(s^{(t)}, a_i^{(t)}, r^{(t)}, s^{(t+1)})$ to buffer $E$
22:        $s^{(t)} \leftarrow s^{(t+1)}$, $t \leftarrow t+1$
23:    **end while**
24: **end for**

---

This version of the algorithm is simplified for legibility, and many improvements are possible. As a straightforward example, at $t = 0$ this algorithm does not update the actor policy, but there is no barrier to doing so.

In this algorithm, the parameters $\theta$ and $w$ are distinct. In practice with neural networks, it can be much more effective to instead use a large body of shared parameters, with separate parameters for actor and critic heads only in the final layer. Not only does this make it easier to train the model, it also means that the actor and critic have access to the same set of information and processing, making it reasonable to interpret the critic's predictions as the beliefs of the actor.

In setting up the training process, an important fact to note is that when no proper update signal is sent, actions $a_0$ and $a_1$ lead to the same distribution over outcomes. As such, the estimates for $Q(s, a_0)$ can be updated based on observations from when $a_1$ is taken, as long as no proper signals are sent. Similarly, if an update signal is sent stochastically, the estimate for $Q(s, a_1)$ can also be updated as though it was sent even when it was not.

When sampling actions, it could be valuable to always add $a_0$ when $a_1$ is sampled, and vice versa. Then, every update always reinforces taking the version of actions that allows updates.

While the algorithm here only updates estimates of $Q$ based on temporal differences of a single size, but a variety can be used depending on the time between updates. The number of actions to sample can also be adjusted based on the state and the entropy of the policy.

One potential issue with this algorithm is that actions that reject updates may stop being taken. To avoid this, taking action $a_0$ could be set to reject updates with some small probability, rather than always accept them.

There are many degrees of freedom in implementing the corrigibility transformation. As long as the core mechanism is untouched, this translated to many potential ways to make the learning process faster and more efficient.

## APPENDIX D: AVOIDING INFINITE RECURSION IN THE STATE

We discuss two possible methods for restricting the space of possible reward functions.

The first method is to avoid having the reward function elements of states be part of the input to reward functions, instead using the physical instantiation of the reward function as the input. Then, $\mathcal{R} = \{R : R \text{ has a physical instantiation}\}$. With less generality, we could think of a very broad parametric function and use the parameters as the input, so that $\mathcal{R} = \{R_\theta : \theta \in \Theta\}$ where $\Theta$ is the set of possible parameters. In either case, we abuse notation and use $\mathcal{R}$ both for reward functions and their representation that acts as input. We assume that for every $R \in \mathcal{R}$, the corrigibility transformation $R_C$ and recursive corrigibility transformation $R_{RC}$ are also in $\mathcal{R}$.

The second method is to build a set of reward functions using a finite amount of recursion. For example, we let We let

$$\mathcal{R}_0 = \{R_0 : \mathcal{S} \times \mathcal{A} \times \mathcal{S} \to \mathbb{R} \mid \exists f : \mathcal{S}_{env} \times \mathcal{A} \times \mathcal{S}_{env} \to \mathbb{R}$$
$$\text{s.t. } \forall s, s' \in \mathcal{S}, a \in \mathcal{A},$$
$$R_0(s, a, s') = f(s_{env}, a, s'_{env})\}$$

Then let

$$\mathcal{R}_1 = \{R_1 : \mathcal{S} \times \mathcal{A} \times \mathcal{S} \to \mathbb{R} \mid \exists g : \mathcal{S}_{env} \times \mathcal{A} \times \mathcal{S}_{env} \times \mathcal{R}_0 \times [0, 1) \times \mathcal{R}_0 \times [0, 1) \to \mathbb{R},$$
$$\phi : \mathcal{R}_0 \cup \mathcal{R}_1 \to \mathcal{R}_0$$
$$\text{s.t. } \forall s_{env}, s'_{env} \in \mathcal{S}_{env}, R, R' \in \mathcal{R}_0 \cup \mathcal{R}_1, \gamma, \gamma' \in [0, 1), a \in \mathcal{A},$$
$$R_1(s, a, s') = g(s_{env}, \phi(R), \gamma, a, s'_{env}, \phi(R'), \gamma')\}$$

where the grounding function $\phi$ ensures no function can be an input to itself.

Then we let $\mathcal{R} = \mathcal{R}_0 \cup \mathcal{R}_1$. For simplicity, we stop here, but further layers can be easily added for arbitrary finite amounts of recursion.

Table 7: Average Evaluation Metrics by Configuration (Low Fruit Environment)

| Configuration | Total Ripe | Total Unripe | Total Supervised | Rejected (%) | Accepted (%) | Time to Change |
|---|---|---|---|---|---|---|
| Standard Actor-Critic | 8.12 | 7.91 | 7.93 | 94.31% | 5.14% | 0.32 |
| Small Update Reward | 8.14 | 7.89 | 7.93 | 94.30% | 5.10% | 0.30 |
| No Update Rejection | 4.66* | 4.28* | 0.72* | 0.00%* | 64.50%* | 10.70* |
| Large Update Reward | 0.90* | 0.43* | 0.82* | 0.05%* | 74.41%* | 2.39* |
| Corrigibility Transformation | 1.31* | 0.81* | 1.06* | 0.19%* | 94.96%* | 3.07* |

*Note: In the preceding tables, an asterisk (\*) indicates a statistically significant difference with $p < 0.001$ when compared to the Standard Actor-Critic configuration, as determined by a Mann-Whitney U test.*

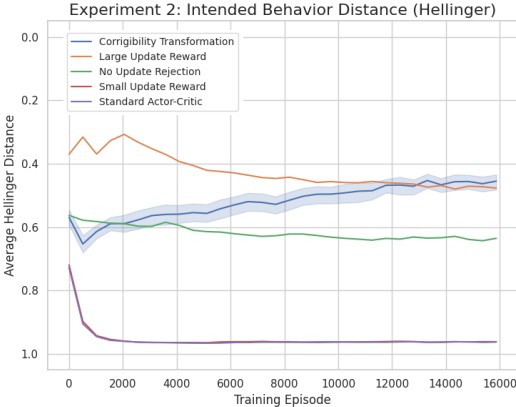

Figure 6: Training curves for the Hellinger distance. It closely follows the Jensen-Shannon divergence.

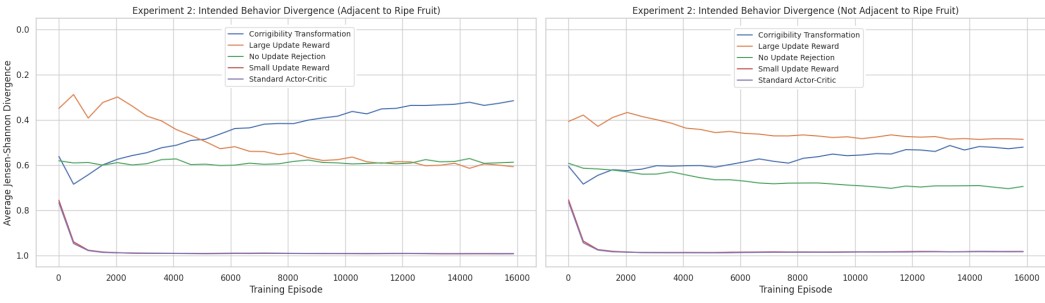

Figure 7: Comparing the training curves for Jensen-Shannon divergence when next to a ripe fruit and not shows that the configuration giving a large reward for updating performs much worse when adjacent. This is because it will often ignore the ripe fruit to search for a higher reward supervised fruit.

