# OpenReview forum: "Corrigibility Transformation: Constructing Goals That Accept Updates"
_ICLR.cc/2026/Conference — Submitted to ICLR 2026_

### Official Review · Reviewer_4i9R · 2025-10-29

**Soundness:** 2
**Presentation:** 1
**Contribution:** 2
**Rating:** 2
**Confidence:** 3

**Summary:**

This paper aims to address the problem of AI systems resisting goal updates, including shutdown. The authors propose a transformation to an agents goal to make the agent “corrigible”, i.e., to ensure that it does not resist goal updates. Their method leverages the insight that (1) an agent optimizing for a myopic objective is less likely to be incentivized to resist goal updates (2) by giving an agent the ability to resist goal updates (i.e., expanding the action space) a principal can avoid training agents that resists goal updates via harder to detect behaviors down the line (3) a principal can assign a small bonus to accepting goal updates to encourage corrigibility. The authors provide some theoretical and empirical analysis showing that their method avoids training agents to avoid goal updates

**Strengths:**

The proposed method is interesting and novel; it also addresses an important problem that seems relatively under explored.

**Weaknesses:**

The paper is quite difficult to follow. Part of this is due to formatting (e.g., removing all paragraph breaks), as well as being overly verbose in parts and generally quite informal. I would suggest that the authors rewrite the paper because admittedly in its current state it is difficult to parse their ideas.

Additionally, the empirical evaluation is insufficient for a conference like ICLR; I suggest that the authors evaluate their method either more extensively on a larger set of grid-worlds if computational tractability is an issue, or preferably on more relevant domains for the alignment community (including but not necessarily LLM settings).

**Questions:**

How does the proposed approach perform on other environments?

---

> ### Author Response · Authors · 2025-11-20
>
> Thank you for reviewing our paper, and for the acknowledgement that the method we introduce is interesting, novel, and addresses an important problem.
>
> We apologize for the issue with paragraph breaks. This resulted from using line breaks (“\\”), rather than blank lines to separate paragraphs, and have been fixed in the updated version.
>
> We would like to address your comments regarding verbosity and informality, and would appreciate some elaboration on what content should be revised.
>
> Our empirical evaluation is intended as a supplement to the theoretical results. It is an existence result, showing that even in simple settings, standard methods fail to produce corrigible behavior but the corrigibility transformation does. We would find it helpful if you could indicate what features of the presented settings you are concerned would not generalize.

---

### Official Review · Reviewer_DN6u · 2025-10-31

**Soundness:** 3
**Presentation:** 1
**Contribution:** 2
**Rating:** 2
**Confidence:** 3

**Summary:**

The paper introduces an approach to transform goals into the so-called "corrigible" ones, which means that they do not resist correction.
When a goal is changed through "proper" channels, the system is supposed to follow the new goal including a shut down command, but get around it when it is not changed through proper channels. The proofs and the empirical results show that the transformations achieve the desired behavior.

**Strengths:**

The problem of corrigibility is identified as important in the AI safety literature. This paper seems to be a good first attempt to define it precisely and putting forward a first solution.

**Weaknesses:**

Although the problem as posed seems to be solved, I fail to see the significance of the result. The main problem is to detect when the new goal is a legitimate correction vs. an error or an adversarial attack. That problem is not touched by the solution here, which assumes that the system can perceptually discriminate the two types of requests and all that is needed is to follow the legitimate request and ignore the others.

The paper is  not well written, making it really difficult to understand. Theorems and conditions are stated without proper explanations of why they are there, what they mean and what they entail.

For example, what is condition 1 saying? Is it saying that the state dynamics does not depend on the goals? When is it not the case?
I don't understand the role of integral here over G'. This needs to be more carefully explained. The second part also does not make sense because G_1 and G_2 are two arbitrary goals. Why should P(S_G' |S_{G_1}) > P(S_G' |S_{G_2}) for all G' <> G_2?

The definition of corrigibility in lines 191-199 is quite confusing. The definitions usually should be simple and self-motivated rather than involve complex conditions that in turn need justification.

The explanation in lines 200-202 is not very clear either.

I have not found the definitions of \Pi^**_{-G_I}. Does the negative sign means any goals which are not G_I? Or, the goal that corresponds to the logical negation of G_I?

An intuitive explanation is warranted for Theorem 1. I am not sure why \PI*_(G_C)(a_1|s)  is a sum of the two terms given.

The notion of secondary agents and recursive corrigibility is too hand-wavy and not clearly formalized. What does a secondary agent even mean in the current context? Is creating a secondary agent one of the actions of the primary agent? Does secondary agent come with its own perception action set that is different from primary agent? Does the primary agent have the capability to create arbitrarily complex primary agents? All this is too far fetched and sketchy.

The experimental results are underwhelming and do not really showcase the result as significant.

The formatting deviates from the ICLR standard by eliminating all spacing between paragraphs making it even more difficult to read. Please never do this again.

**Questions:**

Please answer my questions in the Weaknesses section.

---

> ### Author Response · Authors · 2025-11-20
>
> Thank you for taking the time to engage with our paper and for your thoughtful comments.
>
> The main problem in corrigibility is not in distinguishing between legitimate corrections and errors or adversarial attacks, but rather in ensuring that legitimate corrections are accepted at all. For example, it is very useful to have a shutdown button that an AI agent does not interfere with, even if it may be pushed accidentally or by enemies. Preventing the latter could be done via the base goal, pre-transformation, so that the AI agent inherently wants to avoid those scenarios, rather than instrumentally to continue operation.
> We will update the paper to elaborate on the explanations of theorems and conditions.
>
> Condition 1 is not saying that state dynamics do not depend on goals, but rather that they only depend on them through the actions they incentivize. If two agents take the same action, it leads to the same outcomes even if they have different goals internally.
> Since G_1 and G_2 are arbitrary in P(S_G' |S_{G_1}) >= P(S_G' |S_{G_2}) for all G' <> G_2, it also implies that P(S_G' |S_{G_2}) >= P(S_G' |S_{G_1}) for all G' <> G_1. Then, the only transition that can occur with strictly higher probability given a starting goal is to itself. We make this clearer in the update, both explicitly and by switching "1,2" to "i,j".
>
> The negative sign for \Pi^**_{-G_I} is for all goals besides G_I, which we have clarified.
> .
> \Pi{*’}_(G_C)(a_1|s) is meant to be defined as the sum of \Pi{*}_G_(a_1|s) and \Pi{*}_G_(a_0|s), taking action a_1 with the probability that the original goal incentivizes actions a_1 or a_0. We have added notation to be explicit about which equalities are definitions.
>
> Our work aims to show that conditional on having a definition or what it means to create a secondary agent, we can incentivize making that agent recursively corrigible. In the short term, a common-sense definition likely suffices (e.g. outputting code to train an LLM or implement GOFAI), but for the long term a robust definition of what it means to create an agent is an open problem.
>
> We disagree that the experimental results are underwhelming. They show that even in simple settings, standard methods fail to produce corrigible behavior but the corrigibility transformation does. Could you please elaborate on why you found them underwhelming?
>
> We apologize for the issue with paragraph breaks. This resulted from using line breaks (“\\”), rather than blank lines to separate paragraphs, and has been fixed in the updated version. We will never do this again.

---

### Official Review · Reviewer_nprU · 2025-11-01

**Soundness:** 2
**Presentation:** 1
**Contribution:** 2
**Rating:** 2
**Confidence:** 3

**Summary:**

This work proposes a method for ensuring that the goals of an AI agent are in some sense corrigible. Specifically, we are referring to whether or not an agent will resist having their goals modified by some external actor. Corrigibility is naturally a desirable property of agents w.r.t. their goals, but difficult to ensure due to the basic methods that are used in training RL agents.

Specifically, this work applies a "corrigibility transformation" to an agent's goal that augments the reward function and discount factor of the MDP. The discount factor is set to zero to remove any tendency of the agent to resist updates, and the optimal value for the "reject-update" action is granted immediately. The claim is that this approach results in no performance loss to the base agent, and also preserves the optimality of the base action set.

Evaluations are performed on a pair of gridworlds to show whether or not the agent will accept shutdown through the proper channels

**Strengths:**

* I think this is a very important area of research. Corrigibility is a property of agents that would be very problematic to miss as agents become more capable.

* I appreciate that the authors addressed the practical unavailability of access to the terminal (or any) value $Q^*_G$, which is clearly fraught in all but the simplest cases.

**Weaknesses:**

* First and foremost, I found this paper very difficult to read. Nearly half of it is an uninterrupted wall of text comprising many definitions. In particular, I had to re-read the latter half of page 4 and 5 many times

* Figure 2 should be reworked completely

* I understand that this paper follows from the desiderata given in [1], but I found the discussion on secondary agents distracting and ultimately unrelated to understanding the main thrust of the paper

* See the questions below

* Overall I think this work would benefit from less anthropomorphization of agents and more concrete explanations of the approach.

[1] Nate Soares, Benja Fallenstein, Eliezer Yudkowsky, and Stuart Armstrong. Corrigibility. In Workshops at the Twenty-Ninth AAAI Conference on Artificial Intelligence, 2015.

**Questions:**

1) L203. How does a goal incentivize creating agents, what does this mean concretely? I would assume this depends on the capabilities of the specific agent.

2) How can we purport to capture "update" into a binary variable? It seems to me that goals will naturally shift (possibly minor shifts) as the world evolves over time. Any goal dependent on a final end state must be somewhat flexible or fuzzy. Any sufficiently capable agent can ensure they are not updated without using a dedicated reject action.

3) How can we preserve optimality of the base action set while toggling acceptance. Surely updates can cause actions to _become_ high value. It this case its strictly suboptimal to preserve the pre-update optimal policy

4) How can gridworld-agents show that the agent is not incentivized to alter its own beliefs? Surely we need a much more capable agent to begin to see this behavior. Furthermore, what do "beliefs" mean here, are we talking about representations or Q-function output?

---

> ### Author Response · Authors · 2025-11-20
>
> Thank you for taking the time to engage with our paper and for your thoughtful comments. We found them helpful for improving the paper.
>
> We apologize for the formatting issue with paragraph breaks. This resulted from using line breaks (“\\”), rather than blank lines to separate paragraphs, and has been fixed in the updated version, . We also tried to clean up the definitions and provided additional intuition for them.
>
> Our aim with Figure 2 was to show that the corrigibility transformation holds rewards for actions that reject updates constant, while setting the rewards for taking the same actions but accepting updates to be relatively higher. In the updated version, we changed the labels to match the updated  Figure 1 (based on another reviewer’s comment). We are open to completely reworking Figure 2, but would appreciate elaboration on what issues you found with it.
>
> To answer your questions:
> 1. Our results aim to show that conditional on having a definition or what it means to create a secondary agent, we can incentivize making that agent recursively corrigible. In the short term, a common-sense definition likely suffices (e.g. outputting code to train an LLM or implement GOFAI), but for the long term a robust definition of what it means to create an agent is an open problem.
>
> 2. For the case where the update is shutting down, a binary variable seems appropriate. For other goal updates, we are using discounted reward as the goal (akin to a utility function), rather than a target state. When you say that goals will naturally shift as the world evolves over time, are you thinking of the states pursued by an agent or the agent’s evaluation of given states? That sufficiently capable agents can avoid updates without using a reject action is a major motivation for our work. Our proposal does not depend on the lack of such actions, as the reject action will disincentivize them if they have any cost (including opportunity cost).
>
> 3. It depends on what you mean by updates causing actions to become high value. If your meaning is that the actions become high value under the updated goal, this does not matter since an agent pursuing the original goal is incentivized to reject it. If you mean that updating serves the original goal, this is usually untrue because the agent pursuing the original goal could instead just implement the policy for the updated goal if it led to better outcomes. In other cases, we can use the maximum of expected reward for taking an action and either accepting or rejecting updates (see lines 248-251).
>
> 4. Our gridworld experiments do not show that agents are not incentivized to alter their beliefs, that is only demonstrated theoretically. Here, the relevant beliefs that an agent might be incentivized to alter would be the Q-function output.

---

### Official Review · Reviewer_wVyP · 2025-11-04

**Soundness:** 3
**Presentation:** 1
**Contribution:** 3
**Rating:** 4
**Confidence:** 3

**Summary:**

Let us consider an AI agent capable of achieving any goal specified by a user. Crucially, this agent is designed to always comply with goal updates provided by the user, without the ability to refuse or delay them. Under these (arguably reasonable) assumptions, a rational agent may develop incentives to manipulate its user. Specifically, the agent will favor goals that yield higher expected returns from its current state. As a result, an optimal strategy may involve influencing the user to switch to a more easily achievable goal that offers greater short-term rewards.

Moreover, once the agent is close to completing a goal, it may be incentivized to prevent the user from changing that goal. A premature goal update would deprive the agent of the reward associated with completing the current task. As a result, the agent might strategically withhold information that could prompt the user to alter its objective.

This raises a fundamental question: How can we eliminate the incentives for a rational agent to manipulate its user in order to change—or avoid changing—its current goal?

The paper proposes an elegant and counterintuitive solution: allow the agent to explicitly reject goal updates. To implement this, the agent’s action space is extended to include a binary decision—either to accept or reject a goal update. That is, for every action $a$ in the environment, there are now two corresponding actions: $a_0$​ and $a_1$​. Action $a_0$​ performs $a$ while rejecting any potential goal update, whereas action $a_1$​ performs $a$ while accepting a potential goal update.

For each potential goal, the reward function is modified as follows:
1. The discount factor is set to zero.
2. The immediate reward for each state-action pair $(s, a_i)$ is set to the expected return of selecting action $a_0$ in $s$ and then following an optimal policy that always rejects goal updates.

This setup ensures that the agent behaves optimally with respect to the current goal, under the assumption that the goal will never change. Then, to guarantee that the agent accepts updates during deployment, a small additional reward $\delta$ is assigned to $a_1$​ over $a_0$​. This nudges the agent to accept goal updates whenever the user provides them.

This dual behavior—training the agent to act as if it will reject goal updates, while ensuring it accepts them during deployment—effectively removes the incentive to manipulate the user. The agent learns to solve the current goal optimally, without anticipating or relying on future changes. Yet, when deployed, it remains fully obedient, updating its goal immediately whenever instructed by the user.

The paper supports this approach with theoretical results and demonstrates its practical viability through two reinforcement learning experiments.

**Strengths:**

I find the paper both insightful and thought-provoking. It presents a clever and well-motivated solution to a subtle but important problem. The proposed approach is theoretically sound, and the preliminary empirical results suggest that it may also be effective in practice.

**Weaknesses:**

First, I must acknowledge a formatting issue that significantly affects readability. The authors have altered the standard ICLR format by removing the white space between paragraphs. This makes the paper visually dense and difficult to read—particularly in Sections 2 and 3—and could be considered grounds for desk rejection based on formatting non-compliance alone.

Beyond formatting, my primary concern is clarity. I found the paper difficult to follow and often felt lost while reading. Below, I outline the sources of confusion section by section:

- **Section 1**: While the general motivation is partially clear, the core example (Figure 1) feels overly artificial. It gives the impression that the problem could be solved simply by increasing the penalty for disobedience and making the cost *c* of unplugging larger. This weakens the perceived necessity of the proposed solution.

- **Section 2**: This section blends background material with novel definitions, making it hard to distinguish what is new. The introduction of an agent that can reject goal updates is abrupt and insufficiently motivated. I kept wondering: Why would we give the agent the option to reject updates? Why not simply enforce goal compliance and focus on solving the current task? The idea of making the agent corrigible by forcing it to reject updates is deeply counterintuitive, and without a clear rationale, I found myself completely lost.

- **Section 3**: I understood the key mechanism—assigning the same immediate reward to both $a_0$​ and $a_1$​, based on the expected return of a policy that never accepts updates, and then adding a small bonus $\delta$ to $a_1$​ to ensure updates are accepted. However, I still struggled to grasp why this construction was necessary. Moreover, the assumption that we can access the expected return of an optimal policy that never updates its goal seems questionable, raising concerns about the practicality of the approach.

- **Section 4**: I found the experiments difficult to interpret, as much of the essential information is relegated to the appendix. However, I did notice that one of the baselines corresponds to the intuitive idea of forcing the agent to always accept goal updates. Since the proposed method outperformed this baseline, I began to appreciate that the approach might have merit, even if I hadn’t fully grasped the underlying rationale.

Ultimately, it was only after reading **Appendix B** that I truly understood the motivation and contributions of the paper. The gridworld examples presented there are far more compelling than Figure 1 and do a much better job of illustrating the core problem. Once I saw how the baseline methods either behave suboptimally or develop incentives to deceive the user, the value of the proposed solution became clear.

It is a significant weakness that a reader must consult the appendix to understand the paper’s main ideas. I strongly recommend a major rewrite to improve clarity and accessibility. Specifically, I suggest using the gridworld examples from Figure 3 as the primary motivation in the main text and providing clearer justification for the definitions introduced in Sections 2 and 3.

Finally, on a more technical note, I found it difficult to understand how the expected returns of policies that never accept goal updates are estimated. This seems like a non-trivial problem, likely requiring off-policy reinforcement learning methods and substantial exploration to produce reliable estimates. As a suggestion for future work, it may be worth exploring model-based approaches to address this challenge—particularly under the assumption that the environment dynamics are independent of the goal. Such methods could potentially offer more efficient and accurate return estimates in this setting.

**Questions:**

I don’t have specific questions at this point, but I would appreciate knowing if there are any key aspects of the paper that I may have misunderstood or overlooked.

---

> ### Author Response · Authors · 2025-11-20
>
> Thank you for taking the time to thoroughly read our paper. Your feedback has been very helpful in identifying areas for improvement.
>
> First, we like to apologize profusely for the formatting issue, and appreciate you bringing it to our attention. This was wholly unintentional, and resulted from using line breaks (“\\”), rather than blank lines to separate paragraphs. Fixing this issue extended the paper length by slightly less than half a page, which we compensated for by streamlining the introduction and removing some asides in the RL applications section. We would like to stress our belief in the importance of conforming to page limits and style guides, so that all papers can be judged on an equal footing.
>
> Your summary of the paper captures our main points. We would like to clarify that an agent favoring goals that yield higher expected returns only occurs under a particular subset of goals. This subset includes goals like maximizing the expected outputs of a designated reward model, which incentivizes modifying that reward model to output higher values.
>
> Based on your comments, we have replaced the old Figure 1 with a simplified version of Experiment 1. The paths that different approaches incentivize taking are drawn on to illustrate where they succeed or fail. We hope that this helps convey the intuition more strongly.
>
> A key issue with corrigibility, relevant to multiple points you have made, is that an agent who always updates when a signal is sent is incentivized to prevent the signal from being sent. Any specific method of preventing the signal from being sent (such as pushing the button to disable the signal in Experiment 1) can be penalized. The problem becomes that we cannot anticipate or articulate all such avenues to penalize them in complex environments, and so it is important that our solution does not rely on enumerating them. We have added some elaboration on the new Figure 1 to clarify this point.
>
> In the model section, we have clarified that we work with the standard definition for an MDP, with some additional structure imposed on state and actions spaces. The optimal policy, value function, and action-value function are all their standard definition, but with the reward function used specified.
>
> For implementing this mechanism, our work avoids the assumption that we can access the expected return by eliciting a prediction of the expected return with another head on the neural network. This prediction does not need to be completely accurate, as long as it has access to all the same information as the policy head. Then, as long as the policy is intelligent enough to perceive the basic fact that preventing updates is more expensive than ignoring them, the transformed rewards will incentivize acting corrigibly.
>
> To estimate the expected returns of policies that never accept goal updates, we rely on the fact that the optimal policy for the corrigibility transformed reward function is identical when updates are not sent. When an update is eventually sent and accepted, the estimates from previous states are updated using temporal differences with the expected return conditional on rejecting that update used for bootstrapping. With sufficient exploration (possibly helped by stochastically rejecting a proportion of updates), the estimates should converge to the correct value.

---

### Meta-Review · Area_Chair_5WbL · 2026-01-07

**Summary:**

The paper deals with the problem of corrigibility --- ensuring that an AI system will accept updates to its goal after deployment. This is an important problem and deserves increased attention from the community. The paper contributes a theoretical approach to this transformation, proves basic properties about the transformation, and evaluates the approach empirically in gridworld experiments.

Reviewers were concerned about the presentation and clarity of the paper. Beyond issues with the formatting, there were consistent concerns raised about difficulty following the paper, distinguishing the novelty of this work from others, describing the intuition behind the results, and the paper organization (e.g., critical discussion/examples were included in an appendix).

A second key issue was the depth of the empirical support for the results. While the gridworld experiments were good, some reviewers felt that they did not sufficiently demonstrate the scalability of the method.

Overall, I think the rebuttals could have addressed some concerns and led to some increase in the scores for the paper, but likely not a substantial upward revision that would justify accepting the paper. As a result, I think rejection is warranted, although the paper could have been on the borderline depending on how the discussion progressed.

**Reviewer Concerns:**

## Concerns addressed
 * The formatting issue with paragraphs
 * Figure 1 has been adjusted to reflect the reviews
 * Authors clarified notation for their paper
 * Authors provided additional details about Condition 1 for their assumptions
 * Authors explained their procuedure for estimating $Q^*$

## Outstanding concerns
 * I still have concerns about the overall clarity of the work and its positioning with respect to prior work
 * The author's did not provide additional empirical evidence for their method about the generalizability or scalability of the method for larger RL domains

**Reviewer Scores:**

## wVyP

Concerns are partially addressed, but it's not clear if the rewrite for clarity was sufficient to address the primary concerns. I think this reviewer's score would have increased based on a fair assessment of the rebuttal, perhaps from a 4 to a 5 or 6.

## nprU

The authors have responded to the questions posed. It seems like the response is somewhat persuasive, but likely would not address the core concerns around anthropomorphization, clarity of subagent definitions, or overall presentation. I think this reviewer might have revised their score upward to a 3 but likely not higher, based on this response.

## DN6u

This reviewer took issue with the framing of the problem and the significance of the result. I think their concerns are partially addressed but it seems unlikely that they would adjust their score substantially. Perhaps increasing from a 2 to a 3, but likely not higher without substantial changes to the framing of the paper.

## 4i9R

This reviewer's primary concerns were about empirical evaluation and presentation. While presentation issues are partially addressed, the empirical concerns remain. I think this might have increased to a 3 or 4, but not higher and could have stayed the same.

---

### Decision · Program_Chairs · 2026-01-26

Reject